# Antenatal depression and its relationship with birth outcomes and postnatal depression in Rural India: A longitudinal study

**Deepak** [1]☯*, **Dhananjay W. Bansod**[1‡], **Anke Hoeffler**[2‡]

**1** Department of Public Health & Mortality Studies, International Institute for Population Sciences, Mumbai, India, **2** Department of Politics & Public Administration, University of Konstanz, Konstanz, Germany

☯ These authors contributed equally to this work.
‡ DWB and AH also contributed equally to this work.
* deepakpihal08@gmail.com

## Abstract

### Background

Depression is the most common mental disorder among women of reproductive age, with maternal depression the second leading cause of morbidity worldwide. Despite the serious effects on both the mothers and their newborns, it is still a neglected issue. The purpose of this study is to explore the relationship between antenatal depressive symptoms, birth outcomes, and postnatal depression symptoms.

### Methods

A community-based longitudinal survey was conducted among pregnant women in the rural Chatra district, Jharkhand, India, between April 2023 and February 2024. Participants were selected through multistage random sampling. We followed 246 pregnant women during their antenatal period (Phase I), and followed the same respondents (197 women) after the delivery (Phase II). The Four-Dimensional Symptom Questionnaire (4DSQ), translated into Hindi and validated by back-translation, was employed to assess depressive symptoms with strong internal consistency with Cronbach's alpha = 0.94. Statistical analyses, including χ2 test and multivariate logistic regression, were performed to determine the association between antenatal depression (AND), birth outcomes, and postnatal depression (PND) at a 95% confidence interval, with odds ratios (OR) reported.

### Results

The prevalence of antenatal depression was 56.05%, while postnatal depression was observed for 44.67% of the women. AND was significantly associated with caste category (OR= 4.459), mass media exposure (OR=2.392), family type (OR=3.252), multiple pregnancy (OR= 3.277), and intimate partner violence (OR= 4.424). However,

**Data availability statement:** The dataset used in this study is not publicly accessible due to proprietary, ethical, and confidentiality considerations, as the research forms part of the author's ongoing PhD work. The study protocol was reviewed and approved by the Student Research Ethics Committee (SREC) at the International Institute for Population Sciences. The dataset contains sensitive information on maternal mental health and women's reported experiences of obstetric violence in rural India. Given the sensitive nature of these data and the potential risk of participant re-identification, particularly within small rural communities, the full dataset cannot be placed in the public domain in accordance with SREC guidelines. However, a de-identified version of the dataset, comprising all variables necessary to reproduce the study findings, will be made available to qualified researchers upon reasonable request. Requests for access should be addressed to the Student Research Ethics Committee (SREC) at the International Institute for Population Sciences (IIPS) via email at student.rec@ iipsindia.ac.in. Data will be shared only after the removal of all direct and indirect identifiers, and strictly in accordance with the ethical approvals governing this study.

**Funding:** The author(s) received no specific funding for this work.

**Competing interests:** The authors state that they have no competing interests.

no significant statistical association was found between AND and birth outcome (low birth weight (LBW), preterm birth). Birth outcomes, on the other hand, significantly contributed to PND, LBW (OR= 2.213), birth experience (OR=2.783), and family reaction to birth (OR= 4.323) being key factors. The sex of the child did not show a significant association with PND.

## Conclusion

This study highlights the high prevalence of both AND and PND, as well as the strong association between birth outcomes and PND among women. Early detection and treatment of AND and PND are crucial for improving maternal mental health and infant development.

---

## Introduction

The 2030 Agenda for Sustainable Development sets out targets for healthy lives as well as gender equality in the Sustainable Development Goals (SDGs) 3 and 5, respectively. Maternal mental health is important to achieve these SDGs. The World Health Organization defines maternal mental health as a condition of well-being in which a mother can realize her potential, deal with life's challenges, work productively, and contribute to her society [1–3]. Depression is the most common mental disorder in women during their reproductive years [4], and maternal depression (during pregnancy to 12 months postpartum) is the second leading reason for worldwide morbidity in women [5].

While postnatal depression (PND, up to 12 months postpartum) has received more attention [6], antenatal depression (AND) is also a common health issue [7]. AND is a serious public health concern that is often overlooked in developing countries, including India, both in terms of treatment and research. There are only a few studies on AND and its impact on delivery outcomes and postnatal mental health in India [8,9]. AND is characterized by persistent low mood, anxiety, sleep disturbances, and impaired cognitive function [10]. Previous studies in India found an AND prevalence ranging from 8.7% to 65%, with variations depending on the type of screening instrument used and the study setting [11]. Several risk factors may cause pregnancy-related depression. Some of these include inadequate prenatal care; stressful life events like financial hardship, violence, a history of mental illness, and puerperal complications; a history of pregnancy loss, like prior abortions. Moreover, age, marital status [12], gravidity [13], planned or unplanned pregnancy, a history of stillbirth, a history of premature delivery [14], and the level of social support are additional factors associated with AND [11]. Untreated AND during pregnancy may impact birth outcomes, such as low birth weight, preterm birth, intrauterine growth restriction, and delivery complications [15–17]. In India, premature birth and low birth weight continue to be significant contributors to neonatal and child mortality, with around 3.5 million babies born prematurely and 0.3 million children dying each year before reaching the age of 5, due to complications of adverse birth outcomes [18].

Furthermore, several studies have shown that these adverse birth outcomes significantly increase the risk of PND [19,20]. Recent studies in India reveal the high prevalence of PND ranging between 4% to 48.5% [21,22].

Although a number of separate studies have examined AND, birth outcomes, and PND in India, to our knowledge, there are no studies examining the relationship between AND, birth outcomes, and PND. To address this research gap, the present study investigates the relationship between AND, birth outcomes, and PND among married women in the rural Chatra district of Jharkhand using a longitudinal study design. We examine a rural sample, because although the majority of Indians live in rural areas, most of the existing studies focus on urban areas and clinical settings. In rural areas, there is still limited access to healthcare, posing a significant challenge to maternal (mental) healthcare [23].

## Materials and methods

### Study design

A community-based longitudinal survey was conducted among pregnant women residing in the rural area of Chatra district, Jharkhand, India, between April 2023 (Phase I) and Feb 2024 (Phase II).

The study population is characterized by low literacy rates, a predominance of Scheduled Caste and Scheduled Tribe communities, and limited health infrastructure. Socio-demographic variables—age, education, family type, and caste—were meticulously documented to reflect the broader rural Indian profile. This comprehensive characterization facilitates an informed understanding of the socio-cultural determinants influencing antenatal and postnatal depression within this context [23,24,25].

### Participants

The study population included all pregnant women living in the rural area of Chatra district, excluding those with disability or mental illnesses (e.g., difficulty in hearing or speech impairments, history of dementia with early onset, and brain injuries) to prevent confounding the assessment of antenatal depression (AND) and postnatal depression (PND). Similarly, widows and unmarried women were excluded because of their distinct social and support structures. In Phase II, after childbirth, women who had experienced a miscarriage, stillbirth, or the death of their child by the time of the survey were also excluded to prevent grief-related distress from being misclassified as depressive symptoms. This approach aligns with prior perinatal mental health studies that emphasize stable cohorts for longitudinal tracking [17,26,27].

### Ethics approval and consent to participate

Ethical approval was obtained from the Institutional Review Board of the International Institute for Population Sciences (Student Research Ethics Committee), Mumbai, India, with document number IIPS/ACAD/SREC/D/IO-25/2023.

During the survey, trained research assistants went door-to-door to approach all pregnant women. Eligible women were invited to participate in the study. Participants were given detailed written and verbal information about the research, and interviews were conducted only after obtaining their written informed consent.

Data were collected through one-on-one interviews conducted under the direct supervision of the first author. In order to address the problem of illiteracy and guarantee the same strategy for all participants, interviews were chosen as the data collection process. Each interview lasted about 30 minutes.

To minimize bias and avoid inconsistent explanations of terms, the research assistants explained all terms from the 4DSQ in simple language. They were instructed to follow a guide to provide explanations when needed. Research assistants were evaluated on their ability to use the guide correctly during the pilot and main surveys. Only those who demonstrated competence were involved in the actual data collection.

Data collection for Phase II occurred approximately two months after delivery for each participant. This timing was deliberately chosen based on the cultural practices in rural Jharkhand, India, where recently delivered mothers

 

traditionally observe a 42-day postpartum confinement period (Puerperium). During this time, mothers are typically restricted from social interactions and leaving the house to preserve their health and their infants. By respecting these culturally sensitive practices, we ensured that participants felt comfortable and safe during the postnatal interviews, thereby enhancing data quality and ethical compliance.

## Sample size determination and sampling technique

The sample size (Fig 1) was estimated using Cochran's formula for proportion estimation, $n = (Z^2 \cdot P \cdot (1 - P))/e^2$, where p is the prevalence of common mental disorder, Z is the value from the standard normal distribution that corresponds to the desired confidence level, and e is the desired precision. We assumed the following parameters: d = 0.05, intended accuracy as 0.05, and confidence level of 95%. According to a study performed in rural Haryana in 2021 [28], the incidence of common mental disorders (CMDs) among pregnant women was 15.3%. Based on this, the minimum sample size required was 200, but we surveyed 250 women to allow for attrition. Four interviews were partially completed, so we removed those cases from the analysis, resulting in a sample size of 246 for Phase I (antenatal period).

Of the initial 246 participants, 197 were successfully followed up at Phase II. The primary reasons for attrition were migration (n = 23) and pregnancy loss (miscarriage, stillbirth, or infant death, n = 26).

To assess whether attrition introduced systematic bias, we compared key baseline characteristics—such as age, caste, education, socio-economic indicators, and baseline antenatal depression status—between participants retained in Phase II and those lost to follow-up. Statistical analysis showed no significant differences in these variables, indicating the attrition was unlikely to bias the study findings substantially. Nonetheless, we acknowledge this limitation and caution that the exclusion of women with adverse pregnancy outcomes may have underestimated postnatal depression prevalence in the study population.

A multi-stage sampling technique was employed based on India's administrative structure. The district of Chatra has two subdivisions, Chatra and Simaria, which are divided into administrative blocks and panchayats (the lowest administrative unit). Using Probability Proportional to Size (PPS), 8 panchayats were selected from each block to ensure proportional representation according to the population. Within these panchayats, PPS was applied again to select 50% of the villages, aiming for diversity. The sample frame, developed with the assistance of Health Facility Staff and Accredited Social Health Activist (ASHA worker), documenting details of pregnant women, and a random sampling approach was then employed to select participants. Under the National Rural Health Mission, each village has a trained female ASHA,

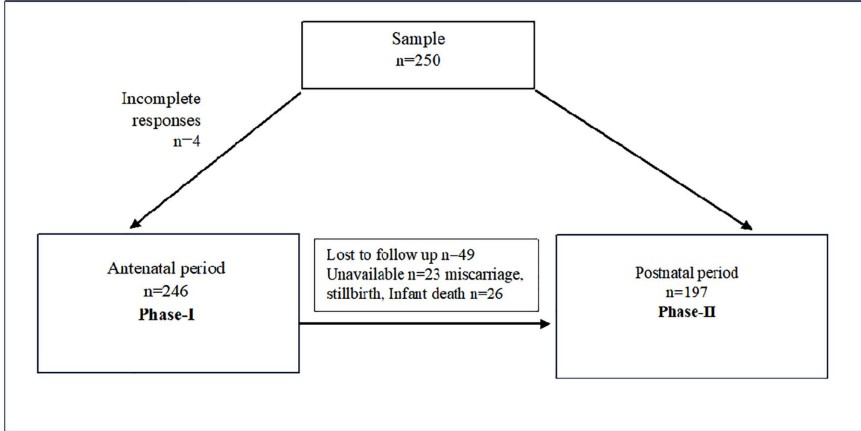

**Fig 1. Sample size distribution.**

appointed by the health department to address the community's healthcare needs. She acts as a link worker, bridging the gap between the public health system and the community. This is of particular importance in communities that lack a fully functional Primary Health Centre (PHC) or a nearby government hospital, as is the case for most villages in the study area.

## Assessment tools

During a single face-to-face interview session, we gathered data on mental health and socio-demographic information. The participants were provided with comprehensive written and verbal information about the research, and the interview was conducted after obtaining their written and oral informed consent.

To assess the prevalence of depression (AND and PND), we employed a validated screening method using the Four-Dimensional Symptom Questionnaire (4DSQ) [29,30]. However, it has not been specifically validated for AND in Indian populations or rural settings to date. Since no Hindi version was available, we translated the instrument and checked for accuracy by back translation to English. The data collection instrument was pretested among 25 pregnant women (10% of the total sample) from a non-sampled rural area in the Chatra district to enhance its validity and reliability. The internal consistency of the 4DSQ tool was evaluated using Cronbach's alpha, resulting in a value of 0.96. Additionally, we conducted an independent check of Cronbach's alpha, which yielded a score of 0.94 for the depression scale in this study.

Maternal mental health was assessed by relying on the depression scale of the 4DSQ. Respondents provided answers to six questions on a 5-point Likert scale (0 = no, 1 = sometimes, 2 = regularly, 3 = often, 4 = very often or constantly), and the answers to each question were categorized into low (0), moderate (1), and high (2–4). By summing these values for all six questions, a depression scale (0–12) was generated, and those with a score of 6 and higher were categorized as depressed [30]. We selected a cutoff score of ≥6 on the 4DSQ depression scale based on prior validation studies demonstrating optimal sensitivity and specificity in perinatal populations [29,30]. The 4DSQ serves as a screening rather than diagnostic instrument, facilitating early identification of depressive symptoms, and agrees that additional validation research targeting antenatal populations in India would be valuable.

Additionally, we utilized the National Family Health Survey 5 (2019−21) [31] questionnaire to address inquiries concerning intimate partner violence, participant socio-demographic profiles, and other relevant characteristics. Participants were also asked about their and their spouses' past or present consumption of alcohol and tobacco products.

## Statistical analysis

The collected data were coded following a comprehensive review for accuracy and completeness. To validate data accuracy, 10% of the sample underwent manual rechecking. The statistical analysis was conducted using STATA software version 17. We first used χ2 tests to examine the correlations between the explanatory variables and the two outcome variables, AND and PND. Variables with significant results (p < 0.05) were included in the logistic regression model, calculating the odds ratio (OR) with a 95% confidence interval. Prior to building the multivariate logistic regression models, we assessed the issue of multicollinearity among explanatory variables using the Variance Inflation Factor (VIF). We found that specific pairs of variables—specifically, education and employment, caste and religion—were highly correlated (VIF consistently >5). To ensure the stability and interpretability of the model estimates, we did not include both variables from each highly correlated pair in the same model. Instead, we selected the more theoretically and contextually relevant variable (e.g., education and caste) for inclusion in each final model.

## Results

Most women were aged 20–24 years (55%), married between 18–21 years (63%), and lived in extended families (70%). Spouses mainly worked in informal jobs (81%), and alcohol use was common (68%). Intimate partner violence was

reported by 72% of women. Most were housewives (81%), received antenatal care (84%), and delivered in government facilities (76%). Hindus (80%), Scheduled Castes (42%), and OBC (41%) made up the majority (see Table 1).

In Phase-I, 246 pregnant women were interviewed. Among them, 139 (56.50%) were found to have a 4SDQ score of ≥6, thus showing a high prevalence of AND (see Supplementary Table S1 in S1 File). **Correlation analysis** showed that only some socio-demographic factors are significantly associated with AND. Depression levels appear to be higher among women from Scheduled Caste and Scheduled Tribe communities, those living in nuclear families, those with precarious household incomes (such as husbands engaged in informal work or unemployed), and those with lower education levels (both respondents and husband's). Age, age at marriage, age at first birth, education, religion, working status, and gestational age were not found to be significantly associated with AND. Detailed results are presented in Table S1 in S1 File.

The analysis of the relationship between obstetric factors and AND showed that neither gestational age (p-value = 0.928) nor a history of pregnancy loss (p-value = 0.896) shows a significant association with AND, indicating that specific pregnancy trimesters do not notably affect the likelihood of experiencing AND. However, the number of pregnancies (primigravida vs. multigravida) is significantly linked to AND (p-value = 0.003) among multigravida women, 84.8% reported high depression levels (see Supplementary Table S2 in S1 File).

**The examination of** spousal behavior and AND suggests that violence experienced during pregnancy, spousal consumption of alcohol, smoking (stimulants), and chewing tobacco are all significantly linked to heightened levels of depression (all p = 0.000) (see Supplementary Table S3 in S1 File).

Table 2 presents a multivariate analysis of AND using the socio-demographic variables commonly examined in the analysis of AND (28) plus variables that are context-specific. Since some of the variables are highly correlated with each other, e.g., caste and religion, education and employment, we only included education and caste in our baseline model (model 1). Education is not statistically significant, but belonging to the Scheduled Tribes or the Scheduled castes increases the risk of experiencing AND (OR=4.459). Alternative models, including employment and religion, are presented in (see Supplementary Table S4 in S1 File). We also control for age, which is insignificant, while living in a nuclear family (OR=3.252) increases the odds of experiencing AND, and mass media exposure (OR=2.392) decreases the odds of experiencing AND. In model 2, we add gravida and find that multigravida women have significantly higher odds of experiencing AND compared to primigravida women (OR=3.277). We then examine the history of pregnancy loss (model 3), which does not show a significant association. Intimate partner violence (IPV), included in model 4, is strongly associated with the experiencing AND (OR=4.424), but alcohol consumption by the spouse (model 5) does not significantly increase the odds of experiencing AND.

A further important question is whether birth outcomes are negatively affected by AND. Birth outcomes in this study are measured as premature birth and low birth weight. Only 7 out of 197 births were premature and thus the number of cases were too small for statistical analysis. The analysis of birth weight showed that babies born to mothers with AND had, on average, no lower birth weights, i.e., AND appears to have no impact on birth weight (see Supplementary Table S5 in S1 File).

We continue our investigation by examining the correlation between PND and birth outcomes. PND is associated with low birth weight (p = 0.02), vaginal delivery (p = 0.03), and family reaction to birth (p = 0.001). Additionally, PND was more common among mothers who felt pressured to have a son but gave birth to another daughter, especially if their previous child was also a girl. (69.57%, P = 0.001). (see Supplementary Table S6 in S1 File).

Table 3 shows the multivariate analysis of PND, as in our examination of AND, we start with our baseline model (column 1), but include AND and low birth weight. Neither age, nor education and caste were statistically significant. Mass media exposure (OR=2.783), living in a nuclear family (OR=2.035), AND (OR=2.050), and low birth weight (OR=2.213) increase the risk of PND. We then add one further additional variable at a time. In model 2, we add the place of delivery, which is not significant. Model 3 shows that a self-reported stressful birth experience significantly increases PND (OR=2.783). Other notable factors, such as no saved money for the delivery, marginally increased the odds of

**Table 1. Socio-demographic characteristics of the respondents.**

| Women Information | | | Spousal Information | | | Family Information | | |
|---|---|---|---|---|---|---|---|---|
| **Age** | **N** | **%** | **Spouse Education level** | **N** | **%** | **Family type** | **N** | **%** |
| <20 | 36 | 14.63 | No Education | 43 | 17.48 | Nuclear family | 75 | 30.49 |
| 20-24 | 135 | 54.88 | Primary | 39 | 15.85 | Extended family | 171 | 69.51 |
| 25-30 | 62 | 25.2 | Middle | 48 | 19.51 | **Type of house** | | |
| 30+ | 13 | 5.28 | Secondary | 54 | 21.95 | Traditional mud house | 97 | 39.43 |
| **Age at Marriage** | | | Higher Secondary & Above | 62 | 25.2 | Semi-permanent house | 75 | 30.49 |
| Below 18 | 73 | 29.67 | **Spouse working status** | | | Modern permanent structure | 74 | 30.08 |
| 18-21 | 155 | 63.01 | Not working | 2 | 0.81 | **Religion** | | |
| Above 21 | 18 | 7.32 | Agriculture | 10 | 4.07 | Hindu | 197 | 80.08 |
| **Age at 1st Birth*** | | | Informal work | 199 | 80.89 | Muslim | 37 | 15.04 |
| Below 18 | 18 | 11.84 | Business | 30 | 12.2 | Christian | 12 | 4.88 |
| 18-21 | 109 | 71.71 | Salaried Job | 4 | 1.63 | **Caste** | | |
| 22-25 | 21 | 13.82 | **Smoking consumption of spouse (incl. stimulants)** | | | SC | 104 | 42.28 |
| 25 & above | 4 | 2.63 | Yes | 104 | 42.28 | ST | 23 | 9.35 |
| **Educational Level** | | | No | 142 | 57.72 | OBC | 102 | 41.46 |
| No Education | 46 | 18.7 | **Alcohol consumption of spouse** | | | General | 17 | 6.91 |
| Primary | 34 | 13.82 | Yes | 167 | 67.89 | | | |
| Middle | 37 | 15.04 | No | 79 | 32.11 | | | |
| Secondary | 64 | 26.02 | **Tobacco consumption of spouse (chewing)** | | | | | |
| Higher Secondary & Above | 65 | 26.42 | Yes | 144 | 58.78 | | | |
| **Working status** | | | No | 102 | 41.22 | | | |
| Housewife | 200 | 81.3 | **IPV during pregnancy** | | | | | |
| Agriculture | 7 | 2.85 | Yes | 178 | 72.36 | | | |
| Informal work | 31 | 12.6 | No | 68 | 27.64 | | | |
| Business | 6 | 2.44 | | | | | | |
| Salaried job | 2 | 0.81 | | | | | | |
| **Living with Spouse** | | | | | | | | |
| Yes | 96 | 39.02 | | | | | | |
| No | 150 | 60.98 | | | | | | |
| **ANC Visits** | | | | | | | | |
| Yes | 206 | 83.74 | | | | | | |
| No | 40 | 16.26 | | | | | | |
| **Birth order*** | | | | | | | | |
| Pregnant with First child | 94 | 38.21 | | | | | | |
| Pregnant with Second child | 84 | 34.15 | | | | | | |
| Pregnant with Third child | 48 | 19.51 | | | | | | |
| Pregnant and having 3 & more children | 20 | 8.13 | | | | | | |
| **Place of last delivery*** | | | | | | | | |
| Government facility | 115 | 75.66 | | | | | | |

*(Continued)*

**Table 1.** (Continued)

| Women Information | | | Spousal Information | | | Family Information | | |
|---|---|---|---|---|---|---|---|---|
| Private facility | 16 | 10.53 | | | | | | |
| Home | 17 | 11.18 | | | | | | |
| Other | 4 | 2.63 | | | | | | |
| **History of pregnancy loss** | | | | | | | | |
| Yes | 68 | 27.64 | | | | | | |
| No | 178 | 72.36 | | | | | | |

*Question asked only those women who have at least one child

experiencing PND in model 4 (OR=2.289), and those females who had more than 4 antenatal visits during their pregnancy were also less likely to experience PND in model 5 (OR=0.561). Multigravida (OR=2.582) and negative family reaction on childbirth (OR=4.323 model 7), show a significant increase in PND. The sex of the child does not show a significant association with PND (model 8).

## Discussion

Our study, conducted among 246 pregnant women in rural India, is distinctive in its comprehensive approach, as we measured and analyzed antenatal depression (AND), birth outcomes, and postnatal depression (PND) while also examining their interrelationships.

Using the 4SDQ instrument, we found that over half of all women (56.5%) experienced AND. This prevalence aligns with previous findings from a systematic review in India, where AND rates varied between 9.2% and 65% across different regions [32]. Socio-demographic factors such as maternal age, age at marriage, age at first birth, and employment status did not show statistically significant associations with AND, a finding consistent with earlier studies in India [8,33]. While no direct correlation was found between a woman's employment status and depression, a notable association emerged between AND and the husband's employment status. This finding is supported by existing research suggesting that having an unemployed spouse increases the likelihood of AND in Asian settings [34–36]. In this community, where husbands typically serve as the primary income earners, their employment status appears to play a crucial role in maternal mental health [34].

Our study reveals a higher prevalence of AND among women in nuclear family setups, a trend consistent with research conducted in Turkey, where the extended family system provides greater emotional and instrumental support compared to nuclear households [37]. Intervention designs should consider family composition, incorporating family-based support where extended networks exist. Studies from rural Africa and Latin America emphasize culturally appropriate mental health programs that leverage familial and community structures to enhance maternal outcomes [38,39]. Given the Indian context, we also examined the relationship between caste categories and AND. Women from the Scheduled Castes and the Scheduled Tribes exhibited higher rates of AND, likely due to the additional social and economic hardships they encounter. These findings align with previous studies conducted in India [24, 37, 40, 41]. Additionally, our study found a strong correlation between a spouse's education level and AND, supporting prior research that has explored this association [42]. A higher level of spousal education may reduce the risk of AND by promoting greater healthcare engagement, as well as providing stronger social and emotional support [40,43,44]. Furthermore, we observed a positive association between the frequency of accessing social media for health information and AND among pregnant women, suggesting that excessive exposure to mass media may negatively impact mental health. Similar correlations have been reported in other studies [45,46].

The experience of intimate partner violence (IPV) is strongly associated with significant mental health distress among pregnant women, evidenced by studies in India and worldwide [26,47]. In Ethiopia, pregnant women with prior experiences

 

**Table 2. Identifying predictors of antenatal depression: multivariate logistic regression analysis.**

| Variables | Model 1 | Model 2 | Model 3 | Model 4 | Model 5 |
|---|---|---|---|---|---|
| **Maternal age** | | | | | |
| **<20®** | | | | | |
| **20-24** | 0.903 | 0.54 | 0.952 | 0.882 | 0.839 |
| | [0.339,2.404] | [0.190,1.532] | [0.353,2.571] | [0.317,2.455] | [0.312,2.257] |
| **25+** | 0.645 | 0.290** | 0.691 | 0.666 | 0.612 |
| | [0.217,1.914] | [0.085,0.987] | [0.228,2.096] | [0.215,2.060] | [0.205,1.828] |
| **Education** | | | | | |
| **At least some education®** | | | | | |
| **No** | 0.73 | 0.622 | 0.748 | 0.506 | 0.643 |
| | [0.234,2.275] | [0.189,2.040] | [0.240,2.333] | [0.151,1.697] | [0.202,2.046] |
| **Mass media exposure** | | | | | |
| **No®** | | | | | |
| **Yes** | 2.392** | 2.899** | 2.479** | 2.179* | 2.311** |
| | [1.062,5.389] | [1.237,6.790] | [1.090,5.637] | [1.053,4.904] | [1.023,5.219] |
| **Caste** | | | | | |
| **General®** | | | | | |
| **SC & ST** | 4.459** | 4.958** | 4.544** | 3.402* | 4.069** |
| | [1.316,15.110] | [1.413,17.400] | [1.340,15.412] | [0.956,12.107] | [1.192,13.890] |
| **OBC** | 2.004 | 2.064 | 2.074 | 1.506 | 2.047 |
| | [0.659,6.095] | [0.660,6.454] | [0.679,6.336] | [0.469,4.831] | [0.670,6.256] |
| **Family type** | | | | | |
| **Extended family®** | | | | | |
| **Nuclear family** | 3.252** | 3.169** | 3.349** | 2.771** | 3.177** |
| | [1.275,8.297] | [1.205,8.337] | [1.309,8.564] | [1.038,7.398] | [1.234,8.176] |
| **Gravida** | | | | | |
| **Primigravida®** | | | | | |
| **Multigravida** | | 3.277*** | | | |
| | | [1.524,7.051] | | | |
| **History of pregnancy loss** | | | | | |
| **No®** | | | | | |
| **Yes** | | | 0.778 | | |
| | | | [0.360,1.682] | | |
| **IPV during pregnancy** | | | | | |
| **No®** | | | | | |
| **Yes** | | | | 4.424*** | |
| | | | | [2.093,9.350] | |
| **Alcohol consumption of spouse** | | | | | |
| **No®** | | | | | |
| **Yes** | | | | | 1.604 |
| | | | | | [0.768,3.354] |

Note: Odds Ratio, reported 95% confidence intervals in brackets * $p<0.10$, ** $p<0.05$, *** $p<0.01$.

OR= Odds Ratio, CI= Confidence Interval, ®= Reference category.

**Table 3.  Identifying predictors of postnatal depression: multivariate logistic regression analysis.**

| Variables | Model 1 | Model 2 | Model 3 | Model 4 | Model 5 | Model 6 | Model 7 | Model 8 |
|---|---|---|---|---|---|---|---|---|
| **Maternal age** | | | | | | | | |
| <20® | | | | | | | | |
| 20-24 | 1.061 | 1.127 | 1.998 | 1.121 | 1.065 | 0.644 | 1.091 | 1.065 |
| | [0.426,2.642] | [0.446,2.848] | [0.711,5.619] | [0.448,2.803] | [0.425,2.667] | [0.233,1.781] | [0.433,2.750] | [0.428,2.652] |
| 25+ | 1.25 | 1.338 | 1.991 | 1.282 | 1.41 | 0.638 | 1.152 | 1.258 |
| | [0.465,3.366] | [0.485,3.688] | [0.669,5.928] | [0.473,3.472] | [0.515,3.857] | [0.202,2.012] | [0.422,3.142] | [0.467,3.390] |
| **Education** | | | | | | | | |
| At least some education® | | | | | | | | |
| No education | 1.149 | 1.089 | 1.248 | 1.058 | 1.012 | 1.143 | 1.227 | 1.123 |
| | [0.408,3.234] | [0.384,3.092] | [0.422,3.686] | [0.368,3.041] | [0.352,2.914] | [0.395,3.307] | [0.421,3.577] | [0.390,3.234] |
| **Mass media exposure** | | | | | | | | |
| No® | | | | | | | | |
| Yes | 2.783** | 2.906** | 2.944** | 2.848** | 2.843** | 3.288** | 2.479* | 2.754** |
| | [1.058,7.324] | [1.089,7.757] | [1.067,8.121] | [1.073,7.563] | [1.074,7.526] | [1.226,8.818] | [1.103,5.901] | [1.041,7.289] |
| **Caste** | | | | | | | | |
| General® | | | | | | | | |
| SC & ST | 2.785 | 2.764 | 5.058* | 2.953 | 2.391 | 3.07 | 2.091 | 2.798 |
| | [0.665,11.665] | [0.655,11.667] | [1.103,25.285] | [0.681,12.799] | [0.562,10.169] | [0.709,13.295] | [0.488,8.963] | [0.667,11.734] |
| OBC | 2.392 | 2.459 | 5.895** | 2.672 | 2.194 | 2.548 | 2.266 | 2.399 |
| | [0.589,9.725] | [0.599,10.100] | [1.103,31.513] | [0.631,11.318] | [0.534,9.006] | [0.608,10.685] | [0.559,9.194] | [0.590,9.758] |
| **Family type** | | | | | | | | |
| Extended family® | | | | | | | | |
| Nuclear family | 2.035* | 2.087* | 1.896 | 1.949* | 2.159** | 1.891 | 2.115* | 2.031* |
| | [1.054,4.124] | [1.085,4.341] | [0.858,4.192] | [1.014,4.013] | [1.014,4.595] | [0.880,4.062] | [1.071,4.112] | [1.041,4.102] |
| **Antenatal depression** | | | | | | | | |
| No® | | | | | | | | |
| Yes | 2.050* | 1.879 | 2.145* | 2.021* | 2.237* | 1.778 | 2.153* | 2.052* |
| | [1.101,4.215] | [0.809,4.369] | [1.120,4.803] | [1.057,4.342] | [1.153,4.943] | [0.763,4.146] | [1.081,4.603] | [1.074,4.304] |
| **Low birth weight** | | | | | | | | |
| No® | | | | | | | | |
| Yes | 2.213* | 2.256* | 2.027 | 2.091* | 2.119* | 2.871** | 2.193* | 2.207* |
| | [1.105,4.910] | [1.122,5.236] | [0.849,4.840] | [1.054,4.867] | [1.065,4.701] | [1.180,6.986] | [1.083,4.905] | [1.074,4.304] |
| **Place of delivery** | | | | | | | | |
| Home® | | | | | | | | |
| Government facility | | 0.371 | | | | | | |
| | | [0.068,2.033] | | | | | | |
| Private facility | | 0.276 | | | | | | |
| | | [0.047,1.608] | | | | | | |
| **Birth experience** | | | | | | | | |
| Normal® | | | | | | | | |
| Stressful | | | 2.783** | | | | | |
| | | | [1.200,6.450] | | | | | |

*(Continued)*

Table 3. (Continued)

| Variables | Model 1 | Model 2 | Model 3 | Model 4 | Model 5 | Model 6 | Model 7 | Model 8 |
|---|---|---|---|---|---|---|---|---|
| Traumatic/sad | | | 2.490* | | | | | |
| | | | [1.054,5.902] | | | | | |
| **Save money for delivery** | | | | | | | | |
| Yes® | | | | | | | | |
| No | | | | 2.289* | | | | |
| | | | | [1.054,5.503] | | | | |
| **Antenatal visits** | | | | | | | | |
| < 4® | | | | | | | | |
| > 4 | | | | | 0.561* | | | |
| | | | | | [0.289,0.980] | | | |
| **Gravida** | | | | | | | | |
| Primigravida® | | | | | | | | |
| Multigravida | | | | | | 2.582** | | |
| | | | | | | [1.175,5.673] | | |
| **Family reaction on birth** | | | | | | | | |
| Good® | | | | | | | | |
| Neutral | | | | | | | 2.301 | |
| | | | | | | | [0.513,10.322] | |
| Bad | | | | | | | 4.323** | |
| | | | | | | | [1.039,17.986] | |
| **Sex of recent child** | | | | | | | | |
| Boy® | | | | | | | | |
| Girl | | | | | | | | 1.069 |
| | | | | | | | | [0.565,2.024] |

Note: Odds ratios (exponentiated coefficients) reported, 95% confidence intervals in brackets* $p < 0.10$, ** $p < 0.05$, *** $p < 0.01$.

OR= Odds Ratio, CI= Confidence Interval, ®= Reference category.

of IPV had a significantly higher risk of AND (Adjusted odds ratio: AOR 4.5, 95% CI: 1.28 to 15.52) compared to those without such a history [48]. Similarly, research in Pakistan showed that women who had endured physical or sexual abuse had a significantly higher odds risk of depression (AOR: 9.25, 95% CI: 6.11 to 14.00) compared to those who had not experienced any form of abuse (AOR: 4.04, 95% CI: 2.81 to 5.81) [34,49]. Women with previous pregnancies were found to be more susceptible to depression than those experiencing pregnancy for the first time. Research in India found that women with multiple pregnancies are more likely to have the risk of depression (AOR: 2.36, 95% CI: 1.19 to 4.66) compared to first-time pregnant women [11]. Research from Nepal and Pakistan shows similar results [50,51]. Multigravida women not only face increased risks associated with pregnancy and childbirth but also grapple with heightened stress as they strive to meet the needs of a newborn while caring for their existing children; social and financial pressures related to caregiving and expectations are mainly associated with depressive symptoms [38, 52, 53]. Our findings also indicate a strong association between partner substance use and increased depression symptoms among the women; this is similar to comparable studies [54–56]. Although a recent systematic review links AND to various adverse birth outcomes such as birth weight, gestational age, and type of delivery [11], evidence from South Asia remains inconclusive. A study of 583 women in Bangalore, a city in southern India, found an association between AND and adverse birth outcomes, particularly low birth weight [57]. In contrast, a

Pakistani study of 763 women in the city of Karachi found no evidence of such an association of AND and adverse birth outcomes [27]. Similarly, our study did not identify a significant link between AND and low birth weight. Although our findings contradict the results from a systematic review [11], the population setting may help to explain these discrepancies. Previous studies in India focused on the poor physical health of mothers and their low literacy rates as explanations of low birth weights [58]. However, several factors may contribute to the lower prevalence of LBW in our sample despite maternal depression: 1) In many rural areas, pregnant women consume traditional, home-cooked, nutrient-rich diets, including pulses, rice, vegetables, and dairy, which may support fetal growth [59]. 2) Rural women often engage in moderate physical activities such as household chores and farming, which can enhance maternal health and circulation. Unlike urban settings, where sedentary lifestyles or work-related stress are common, rural women may experience better metabolic adaptation during pregnancy, promoting healthy fetal development [60]. 3) Smoking, alcohol consumption, and processed food intake were found to be nearly negligible in our sample, which could positively influence fetal growth [61,62]. 4) The tribal and rural populations of Jharkhand may have developed better maternal-fetal adaptations over generations, potentially reducing the incidence of LBW [63]. Our finding, i.e., the insignificant association between AND and birth outcomes, could thus be due to cultural and nutritional factors promoting foetal growth in this rural population despite maternal depression. However, other factors could also be influencing our analysis, such as limited sample size, reduced power, and the timing of assessment post-delivery. The heterogeneity of the association between AND and birth outcomes has been noted in South Asian contexts, suggesting the need for nuanced interpretation [57,27,59].

Out of the 197 women who experienced a live birth, 44.67% suffered from PND. There is overwhelming evidence that adverse birth outcomes are a predictive factor for PND [64,65]. A study conducted in Chhattisgarh state of India reported that most deliveries occur in public hospitals, where inadequate resources and a negative hospital environment contribute to depressive symptoms [66]. However, our study did not find an association between PND and the place of birth (home, public, or private hospital). Various studies indicated that negative family reactions to childbirth can significantly contribute to PND among women in Indian society. Our research found that those women who have experienced negative family reactions to childbirth are more likely to experience PND. These findings highlight the importance of family support and cultural sensitivity in managing PND, especially in contexts where traditional and familial expectations around childbirth play an important role in the mother's mental health outcomes [9,67].

In India, a strong preference for sons has been linked to PND in previous research [22]. However, our study did not find a significant association between the sex of the newborn and PND. This could be due to the support provided by the extended family, though further research is needed to confirm this.

Additionally, we found a positive association between AND and PND, consistent with a study from Madhya Pradesh that identified AND as a significant predictor of PND. This emphasizes the need for early screening and intervention [68]. Previous studies found that socioeconomic factors, such as education level [69] and limited community support, can further increase the risk of PND [9,25]. However, our study did not find a significant correlation between PND and education status. This may be due to the poor quality of schooling in rural areas, where additional years of education do not necessarily increase understanding and knowledge [70]. Findings from other Indian studies, such as one in rural Bihar, point to the same mechanism, suggesting that economic hardship persists regardless of educational attainment, and poor financial status was identified as a more critical predictor of perinatal depression [71]. Research also indicates that, despite higher education levels, stigma and a lack of awareness about mental health issues prevent women from seeking help, contributing to the high prevalence of PND [22]. Furthermore, studies show that multiple pregnancies and early marriage—both common in rural India—heighten women's vulnerability to stress and depression, irrespective of their education level [39].

## Limitations of the study

Our study was conducted in a rural setting, where levels of poverty are higher than in urban areas. Consequently, our findings may not be generalizable to urban pregnant women. Unfortunately, during the study period, no follow-up visits were

conducted for women who had experienced abortions or stillbirths, preventing us from exploring their association with PND. Recall bias may affect subjective measures such as birth experience, potentially influencing reported associations. Moreover, the exclusion of women experiencing adverse birth outcomes (miscarriages, stillbirths) during Phase II may have resulted in underestimation of postnatal depression prevalence, as previous research shows these outcomes are strongly associated with a higher risk of depression [19,27]. Additionally, we did not consider other factors, such as wealth status, women's empowerment, and the quality of social support.

The 4DSQ tool used for screening AND and PND serves as a screening instrument rather than a confirmatory diagnostic test. Furthermore, given the small sample size, the wide confidence intervals seen for some variables (like caste) probably indicate limited statistical power. Some participants may have also underreported their depressive symptoms, a challenge noted in previous research [3].

The study setting follows a patrilocal tradition, and the presence of mothers-in-law during data collection posed an additional challenge. Although efforts were made to ensure confidentiality by requesting their absence, some remained present, potentially influencing participants' responses. Similar issues have been documented in other studies, where the presence of family members, particularly mothers-in-law, restricted open discussion, particularly when responding to delicate questions about intimate partner violence and family dynamics, which could have resulted in underreporting because of social desirability bias [27].

## Conclusions

Mental health and gender equality, central to SDGs 3 and 5, remain under-addressed in India's primary healthcare system. This study from rural Chatra district, Jharkhand, reveals a high prevalence of antenatal and postnatal depression and strong links with adverse birth outcomes. Key predictors of antenatal depression include intimate partner violence, higher gravidity, caste disadvantage, nuclear family living, and mass media exposure, while negative family reactions, adverse birth experiences, and multiple deliveries increase postnatal depression risk. Women from Scheduled Castes and Tribes face disproportionate vulnerability due to intersecting structural, cultural, and socioeconomic barriers, including stigma, poverty, limited services, gender-based violence, and restricted autonomy. Maternal healthcare must ensure that every woman is treated with dignity and equity, irrespective of caste, socioeconomic status, or family context.

The findings highlight the urgent need to integrate culturally sensitive mental health screening, IPV identification, and family-centred counselling into routine antenatal and postnatal care, particularly through community platforms such as ASHA-led services. Strengthening referral linkages between primary and specialized mental health care is essential to prevent intergenerational impacts. Leveraging existing maternal health programs to deliver scalable, community-based mental health interventions can substantially improve outcomes for mothers and children in rural India.

## Supporting information

**S1 File. S1 Table. Bivariate analysis of socio-demographic characteristics associated with antenatal depression (AND).** S2 Table. Association of obstetric factors with AND. S3 Table. Association of spousal factors with AND. S4 Table. Identifying predictors of AND: logistic regression analysis. S5 Table. Association of Birth outcomes with AND. S6 Table. Association of birth outcomes with postnatal depression.
(DOCX)

**S2 File. Antenatal Phase questionnaire1.**
(PDF)

**S3 File. Postnatal_Phase_questionnaire1.**
(PDF)

**S4 File. Inclusivity-in-global-research-questionnaire.**
(DOCX)

## Acknowledgments

We would like to thank all the women who took part in the survey and the enumerators. Special thanks to Abena Yalley for her support and interest throughout the project. All remaining errors are our own. Deepak gratefully acknowledges the discussions with colleagues at the Zukunftskolleg at the University of Konstanz, Germany. We would like to thank Shashanka Ashili and Sunandita Das for helping with proofreading the manuscript.

## Author contributions

**Conceptualization:** Dhananjay W. Bansod, Anke Hoeffler.

**Data curation:** Deepak -.

**Formal analysis:** Deepak -.

**Investigation:** Dhananjay W. Bansod.

**Methodology:** Dhananjay W. Bansod, Anke Hoeffler.

**Supervision:** Dhananjay W. Bansod, Anke Hoeffler.

**Writing – original draft:** Deepak -, Anke Hoeffler.

**Writing – review & editing:** Dhananjay W. Bansod, Anke Hoeffler.

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
