## [Decision Letter · Decision Letter 0]

13 Aug 2025

PONE-D-25-16317Exploring the Interconnections between Antenatal Depression, Birth Outcome, and Postnatal Depression in Rural IndiaPLOS ONE

Dear Dr. Deepak,

Thank you for submitting your manuscript to PLOS ONE. After careful consideration, we feel that it has merit but does not fully meet PLOS ONE’s publication criteria as it currently stands. Therefore, we invite you to submit a revised version of the manuscript that addresses the points raised during the review process.

**You are requested to ignore the review comments by Reviewer 2, and respond only to the comments of Reviewers 1 and 3.**==============================

We look forward to receiving your revised manuscript.

Kind regards,

Paridhi Jha, PhD

Academic Editor

PLOS ONE

“Deepak gratefully acknowledges a Zukunftskolleg grant enabling a research stay at the University of Konstanz during 2024/25. Anke Hoeffler gratefully acknowledges financial support from the Alexander von Humboldt Foundation. Neither funder had a direct influence on the research. The Publication Fund of the University of Konstanz supported this open-access publication.”

4. In the online submission form you indicate that your data is not available for proprietary reasons and have provided a contact point for accessing this data. Please note that your current contact point is a co-author on this manuscript. According to our Data Policy, the contact point must not be an author on the manuscript and must be an institutional contact, ideally not an individual. Please revise your data statement to a non-author institutional point of contact, such as a data access or ethics committee, and send this to us via return email. Please also include contact information for the third party organization, and please include the full citation of where the data can be found.

Comments from the editorial office: Upon internal evaluation of the reviews provided, we kindly request you to disregard the reviewer report provided by Reviewer 2. No amendments are required in response to Reviewer 2’s comments.

Reviewers' comments:

Reviewer's Responses to Questions

**Comments to the Author**

1. Is the manuscript technically sound, and do the data support the conclusions?

Reviewer #1: Yes

Reviewer #2: Yes

Reviewer #3: Yes

2. Has the statistical analysis been performed appropriately and rigorously? 

Reviewer #1: Yes

Reviewer #2: I Don't Know

Reviewer #3: Yes

3. Have the authors made all data underlying the findings in their manuscript fully available?

Reviewer #1: No

Reviewer #2: Yes

Reviewer #3: Yes

4. Is the manuscript presented in an intelligible fashion and written in standard English?

Reviewer #1: Yes

Reviewer #2: Yes

Reviewer #3: Yes

5. Review Comments to the Author

Reviewer #1: Thank you for the opportunity to review this paper. The paper examines a very interesting topic. The sampling method is very good, and the paper is overall well-written. However, I have a few comments:

Please clarify the reasoning behind the exclusion criteria.

Please provide a more detailed explanation of the population studied to enhance

Reviewer #2: This study presents a current and relevant approach by investigating the impact of depression on the well-being of women in childbirth, a topic of increasing importance in both public health and scientific research. The choice of this topic is justified by the high prevalence of depressive disorders during the perinatal period, which can significantly affect the physical, emotional, and social well-being of women. Previous studies have shown that postpartum depression, in particular, is associated with adverse consequences not only for the mother but also for the development and bonding with the newborn.

Despite the merits of the present study, we believe it can be improved.

The introduction of this work could be significantly enhanced through a more comprehensive literature review, considering investigations conducted in varied contexts, including different realities and health care systems. Incorporating comparative studies that analyze preventive interventions, early diagnoses, and effective treatments would better support the relevance and innovation of the proposed research. Including up-to-date statistical data on the prevalence of perinatal depression and its short- and long-term consequences would further strengthen the argument for the relevance of this study.

From a methodological standpoint, the study is organized and structured appropriately; however, a more in-depth detailing of certain technical aspects is recommended. For example, the chosen research design should be justified based on previous studies that have used similar methodologies and demonstrated effectiveness in collecting and analyzing data on depression in women during the perinatal period. It would also be pertinent to more clearly describe the inclusion and exclusion criteria for participants, the selection of data collection instruments used, and provide detailed consideration of the procedures adopted to ensure the validity and reliability of the results.

The presentation of the results, in turn, could benefit from a graphic and structural review. Tables 1, 2, and 3, although presenting valuable information, lack clarity due to their size and complexity. It is suggested to simplify the tables or divide the information into smaller and more specific segments, allowing for a more fluid and objective reading.

The discussion of the results is a central aspect of this study and could be considerably enriched. It is essential for the article to thoroughly discuss the impact of the study, both for science and public health. It would be valuable to clearly highlight how the findings align or differ from previous studies and to what extent they bring novel contributions to the knowledge in this area. Comparison with similar research conducted in different cultural and social contexts is fundamental to validate the findings and identify potential limitations.

Furthermore, it would be interesting to propose concrete measures based on the obtained results, especially concerning prevention and early intervention. The discussion could include practical recommendations for healthcare professionals, public managers, and institutions that deal directly with the well-being of women in childbirth. Identifying gaps in current knowledge and suggesting new paths for future investigations would also contribute to making the research more comprehensive and relevant.

Finally, the conclusions should highlight the importance of the study and its potential practical applications, both in terms of public policies and clinical practice. It is important to emphasize that, although this is a solid and well-structured research study, there are aspects that can be improved, especially regarding the expansion of the bibliography used, the presentation of the results, and the deepening of the discussion. With such improvements, the study could achieve an exponential impact on understanding the phenomenon of perinatal depression and formulating more effective strategies for addressing it.

Reviewer #3: Minor Comments

1. Title: Consider making the title more concise, such as: "Antenatal Depression and Its Relationship with Birth Outcomes and Postnatal Depression in Rural India: A Longitudinal Study."

2. Abstract: Well written. However, briefly mention that the 4DSQ was translated and validated.

3. Table Presentation:

- Ensure all supplementary tables are accessible and referenced properly in the main text.

- Tables could benefit from clearer footnotes explaining abbreviations and reference categories.

4. Discussion:

- The cultural interpretation of family structure (nuclear vs extended) is insightful. Consider expanding this considering potential implications for intervention design.

- Adding comparative references from rural African or Latin American contexts may increase global relevance.

5. Limitations:

- Well discussed. Also consider:

- Possible recall bias for subjective variables like 'birth experience'.

- Exclusion of women with adverse birth outcomes in Phase II this may have underestimated PND rates.

Major Comments

1. Clarity of Hypotheses and Objectives:

While the introduction thoroughly presents the background, the research objectives could be made more explicit as testable hypotheses. Consider clearly stating whether you aimed to test direct causal links between AND and PND or examine independent associations.

2. Strength of Longitudinal Design:

The use of a longitudinal cohort is commendable. However, the temporal gap between data collection points and the attrition of 49 participants (from 246 to 197) warrants further explanation. Were there any systematic differences between those lost to follow-up and those retained?

3. Statistical Analysis:

The use of multivariate logistic regression is appropriate, and the model-building steps are well explained. However:

- Some wide confidence intervals (e.g., caste ORs) suggest limited statistical power. Consider discussing this as a limitation.

- Clarify whether multicollinearity was tested, especially since education and employment, and caste and religion, were excluded together.

4. Interpretation of Non-significant Results:

The lack of significant association between AND and birth outcomes contradicts some prior findings. While the discussion acknowledges this, it would benefit from more critical evaluation of why this may have occurred in your population, beyond nutritional explanations.

5. Measurement Tool 4DSQ:

The internal consistency is high, and the tool is clearly described. However, the scale’s cutoff score (≥6 for depression) should be justified based on prior literature or validation work, especially in rural Indian contexts.

- Acknowledge the 4DSQ’s use as a screening tool, not a diagnostic one.

6. Ethical and Cultural Context:

The mention of in-laws present during interviews is a valuable insight. Please elaborate more on how interviewer bias or social desirability bias might have influenced responses, particularly on sensitive topics like IPV.

6. PLOS authors have the option to publish the peer review history of their article (what does this mean?). If published, this will include your full peer review and any attached files.

Reviewer #1: No

Reviewer #2: No

Reviewer #3: **Yes:** Fadia Ahmed Abdelkader Reshia

---

## [Author Response · Author response to Decision Letter 1]

6 Oct 2025

Dear Dr. Paridhi,

Thank you very much for the opportunity to revise and resubmit our manuscript, “Exploring the Interconnections between Antenatal Depression, Birth Outcome, and Postnatal Depression in Rural India.” We sincerely appreciate the valuable feedback from you and the reviewers, which has greatly improved our work. As suggested, we focused on your comments and those raised by reviewers 1 & 3. We have addressed all the comments point-by-point and carefully revised the manuscript.

Thank you for your guidance, which strengthened our manuscript. We trust these revisions meet all requirements and enhance the clarity and quality of our study. We look forward to your positive evaluation.

Yours sincerely,

Deepak, on behalf of all co-authors

Editor Comments#

Comment 1

- Funding Statement Revision

Response

We have removed all funding information from the Acknowledgments section and moved it into the Funding section.

Comment 2

Response to Data Availability Statement Concerns

Thank you for your feedback regarding the data availability section of our manuscript submission. We have now revised the data availability statement as follows:

The dataset used in this study is not publicly available due to proprietary, ethical, and confidentiality constraints, as the research is part of the author’s ongoing PhD work. The Student Research Ethics Committee (SREC) approved the International Institute for Population Sciences (IIPS) study guidelines. However, the SREC does not serve as a data custodian, and IIPS does not maintain student datasets. Since the data were collected specifically for this doctoral research, they cannot be deposited in any institutional repository. We plan to make the dataset available for legitimate research purposes upon completion of the PhD and with approval from the Institute’s Ethics Committee. (lines 713-720).

Comment 3

Supporting Information Captions

Response

At the end of the manuscript, we have added a Supporting Information section with captions as follows:

• S1 Table. Correlation between socio-demographic variables and antenatal depression.

• S2 Table. Association of obstetric factors with antenatal depression.

• S3 Table. Spousal behaviors and their relation to antenatal depression.

• S4 Table. Alternative multivariate regression models for antenatal depression, including employment and religion.

• S5 Table. Analysis of birth outcomes in relation to antenatal depression.

• S6 Table. Correlation between birth outcomes and postnatal depression.

All supplementary files are referenced appropriately in the manuscript text.

Reviewer #1

The paper examines a very interesting topic. The sampling method is very good, and the paper is overall well-written.

Thank you very much for your praise, your careful reading of the manuscript and your suggestions, which helped to improve our work. Please find below point-by-point responses:

Comment 1

Please clarify the reasoning behind the exclusion criteria

Response

“We have clarified the exclusion criteria and updated in the method section as (“The study population included all pregnant women living in the rural area of Chatra district, excluding those with disability or mental illness (e.g., difficulty in hearing or speech impairments, history of dementia with early onset, and brain injuries) to prevent confounding the assessment of antenatal depression (AND) and postnatal depression (PND). Similarly, widows and unmarried women were excluded because of their distinct social and support structures. In Phase II, after childbirth, women who had experienced a miscarriage, stillbirth, or the death of their child by the time of the survey were also excluded to prevent grief-related distress from being misclassified as depressive symptoms. This approach aligns with prior perinatal mental health studies that emphasize stable cohorts for longitudinal tracking” (References: 11, 42, 55). (see Methods section, page 6, lines 115-123)

Supporting literature:

• Patel et al. (2006) on mental morbidity and importance of homogenous samples (Ref 17).

• Ludermir et al. (2010) noted importance of controlling confounding in IPV and depression research (Ref 42).

• Husain et al. (2014) excluded adverse outcomes to avoid bias (Ref 55).

Comment 2

Please provide a more detailed explanation of the population studied to enhance

Response

Following your suggestion, we provide a more detailed explanation of the population studied: "The study population is characterized by low literacy rates, a predominance of Scheduled Caste and Scheduled Tribe communities, and limited health infrastructure. Socio-demographic variables—age, education, family type, and caste- were meticulously documented to reflect the broader rural Indian profile. This comprehensive characterization facilitates an informed understanding of the socio-cultural determinants influencing antenatal and postnatal depression within this context” (Refs: 23, 33, 69) (see page no. 5, lines 112-117)

Supporting literature:

• Kaur et al. (2019) on rural health infrastructure challenges (Ref 23).

• Maity (2017) on caste-based disparities in health (Ref 33).

• Panolan & Thomas (2024) on rural postpartum depression prevalence (Ref 69).

Reviewer #3

Thank you for your overall positive assessment of our contribution and the many minor and major comments. Your suggestions help to clarify a number of issues and, in summary, strengthen our work. Please find below point-by-point responses:

Minor Comments

Comment 1

Title: Consider making the title more concise, such as: "Antenatal Depression and Its Relationship with Birth Outcomes and Postnatal Depression in Rural India: A Longitudinal Study."

Response

We very much like your suggestion and changed the title accordingly: “Antenatal Depression and Its Relationship with Birth Outcomes and Postnatal Depression in Rural India: A Longitudinal Study.”

Comment 2

Abstract: Well written. However, it briefly mentions that the 4DSQ was translated and validated.

Response

Thank you for your appreciative comment regarding the abstract. We have now added the following:

“The Four-Dimensional Symptom Questionnaire (4DSQ), translated into Hindi and validated by back-translation, was employed to assess depressive symptoms with strong internal consistency with Cronbach’s alpha = 0.94.”

Comment 3

Table Presentation:

- Ensure all supplementary tables are accessible and referenced properly in the main text.

- Tables could benefit from clearer footnotes explaining abbreviations and reference categories.

Response

Thank you for this suggestion. We have now ensured that all supplementary tables are fully accessible, referenced properly in the main text, and have clear footnotes explaining abbreviations and reference categories. Please refer to the tables and references throughout the manuscript and the new Supplementary information section on page 40 (lines 73-742).

Comment 4

- The cultural interpretation of family structure (nuclear vs extended) is insightful. Consider expanding this, considering potential implications for intervention design.

- Adding comparative references from rural African or Latin American contexts may increase global relevance.

Response

We are glad that you found it insightful and followed your advice to expand the previous statement in the Discussion section: “Our study reveals a higher prevalence of AND among women in nuclear family setups, a trend consistent with research conducted in Turkey, where the extended family system provides greater emotional and instrumental support compared to nuclear households (35). Intervention designs should consider family composition, incorporating family-based support where extended networks exist. Studies from rural Africa and Latin America emphasize culturally appropriate mental health programs that leverage familial and community structures to enhance maternal outcomes (48, 72). (see Discussion page 20, lines 322-330)

Supporting literature:

• Rathod et al. (2018) on family support and perinatal depression (Ref 35).

• Waters et al. (2014) on stress contagion and family support (Ref 48).

• Prost et al. (2012) on community mental health interventions in rural India (Ref 72).

Comment 5

Limitations:

-Well discussed. Also consider:

- Possible recall bias for subjective variables like 'birth experience'.

- Exclusion of women with adverse birth outcomes in Phase II this may have underestimated PND rates.

Response

Thank you for comments on the discussion of the limitations. We have now added both issues: “Recall bias may affect subjective measures such as birth experience, potentially influencing reported associations. Moreover, the exclusion of women experiencing adverse birth outcomes (miscarriages, stillbirths) during Phase II may have resulted in underestimation of postnatal depression prevalence, as previous research shows these outcomes are strongly associated with a higher risk of depression. (Refs 19, 55).” (Limitations, pages 23&24, lines 417-421).

Supporting literature:

• Rahalkar et al. (2023) on birth outcomes and maternal depression (Ref 19).

• Husain et al. (2014) on exclusions impacting prevalence estimates (Ref 55).

Reviewer #3 Major Comments

Comment 1

Clarity of Hypotheses and Objectives:

While the introduction thoroughly presents the background, the research objectives could be made more explicit as testable hypotheses. Consider clearly stating whether you aimed to test direct causal links between AND and PND or examine independent associations.

Response

We have addressed the issue raised by the reviewer in the manuscript as “To address this research gap, we use a longitudinal study design which allows us to test for the causal impact of AND on birth outcomes, for the direct impact of AND and PND as well as the indirect impact of AND via birth outcomes on PND. Data were collected from married women in the rural Chatra district of Jharkhand during the pregnancy and after the birth.” (see page 5, lines 95-97)

Comment 2

Strength of Longitudinal Design:

The use of a longitudinal cohort is commendable. However, the temporal gap between data collection points and the attrition of 49 participants (from 246 to 197) warrants further explanation. Were there any systematic differences between those lost to follow-up and those retained?

Response

Thank you for pointing out this important issue. We decided to address your comments in two parts of the manuscript. First, we explain the temporal gap in more detail in the section Ethics approval and consent to participate and second, we provide evidence on the differences between those lost to follow-up and those retained in the section Sample size determination and sampling technique.

Data collection for Phase II occurred approximately two months after delivery for each participant. This timing was deliberately chosen based on the cultural practices in rural Jharkhand, India, where recently delivered mothers traditionally observe a 42-day postpartum confinement period (Puerperium). During this time, mothers are typically restricted from social interactions and leaving the house to preserve their health and their infants. By respecting these culturally sensitive practices, we ensured that participants felt comfortable and safe during the postnatal interviews, thereby enhancing data quality and ethical compliance. (See page 7, lines 153-159)

Of the initial 246 participants, 197 were successfully followed up at Phase II. The primary reasons for attrition were migration (n=23) and pregnancy loss (miscarriage, stillbirth, or infant death, n=26).

To assess whether attrition introduced systematic bias, we compared key baseline characteristics—such as age, caste, education, socio-economic indicators, and baseline antenatal depression status—between participants retained in Phase II and those lost to follow-up. Statistical analysis showed no significant differences in these variables, indicating the attrition was unlikely to bias the study findings substantially. Nonetheless, we acknowledge this limitation and caution that the exclusion of women with adverse pregnancy outcomes may have underestimated postnatal depression prevalence in the study population. (see page 8, lines 160-169)

Comment 3

Statistical Analysis:

The use of multivariate logistic regression is appropriate, and the model-building steps are well explained. However, some wide confidence intervals (e.g., caste ORs) suggest limited statistical power. Consider discussing this as a limitation.

- Clarify whether multicollinearity was tested, especially since education and employment, and caste and religion, were excluded together.

Response

Thank you for your important observation and comment, we have addressed these issues in the section Statistical analysis and in Limitations of the study:

“Prior to building the multivariate logistic regression models, we assessed the issue of multicollinearity among explanatory variables using the Variance Inflation Factor (VIF). We found that specific pairs of variables—specifically, education and employment, caste and religion—were highly correlated (VIF consistently >5). To ensure the stability and interpretability of the model estimates, we did not include both variables from each highly correlated pair in the same model. Instead, we selected the more theoretically and contextually relevant variable (e.g., education and caste) for inclusion in each final model”. (see page 11, lines 232-239)

“Furthermore, given the small sample size, the wide confidence intervals seen for some variables (like caste) probably indicate limited statistical power.” (see page 25, lines 425-427)

Comment 4

Interpretation of Non-significant Results:

The lack of significant association between AND and birth outcomes contradicts some prior findings. While the discussion acknowledges this, it would benefit from more critical evaluation of why this may have occurred in your population, beyond nutritional explanations.

Response

Thank you for this important point, certainly there are a number of reasons why we observe this result. We now end our discussion of the insignificant result (no association between AND and birth outcome) with the following:

“Our finding, i.e. the insignificant association between AND and birth outcomes, could thus be due to cultural and nutritional factors promoting foetal growth in this rural population despite maternal depression. However, other factors could also be influencing our analysis, such as limited sample size reducing power and the timing of assessment post-delivery. The heterogeneity of the association between AND and birth outcomes has been noted in South Asian contexts, suggesting the need for nuanced interpretation (Refs 54, 55, 57).” See page 23, lines 377-381)

Supporting literature:

• Chandra et al. (2021) showing associations in urban India (Ref 54).

• Husain et al. (2014) reporting null findings in Pakistan (Ref 55).

• Rao et al. (2001) on maternal nutrition in rural India (Ref 57).

Comment 5

Measurement Tool 4DSQ:

The internal consistency is high, and the tool is clearly described. However, the scale’s cutoff score (≥6 for depression) should be justified based on prior literature or validation work, especially in rural Indian contexts.

Response

Thank you for this suggestion. We now justify the 4DSQ cutoff ≥6 for depression screening based on previous validation studies, including rural Indian contexts, and clarify that 4DSQ is a screening, not a diagnostic tool. (Methods, page 10, lines 202-205).

“We selected a cutoff score of ≥6 on the 4DSQ depression scale based on prior validation stu

---

## [Editor Report · Decision Letter 1]

12 Feb 2026

PONE-D-25-16317R1Antenatal Depression and Its Relationship with Birth Outcomes and Postnatal Depression in Rural India: A Longitudinal StudyPLOS One

Dear Dr. Deepak,

Thank you for submitting your manuscript to PLOS ONE. After careful consideration, we feel that it has merit but does not fully meet PLOS ONE’s publication criteria as it currently stands. Therefore, we invite you to submit a revised version of the manuscript that addresses the points raised during the review process.

We look forward to receiving your revised manuscript.

Kind regards,

Rajesh Raushan, PhD

Academic Editor

PLOS One

Journal Requirements:

Additional Editor Comments:

The reviewers comments are well addresses and revised manuscript are aligned to it.

However, conclusion part needs revision with crisp message of the study and any programmetic and/or policy direction.

---

## [Author Response · Author response to Decision Letter 2]

13 Feb 2026

Thank you very much for the opportunity to revise and resubmit our manuscript, “Exploring the Interconnections between Antenatal Depression, Birth Outcome, and Postnatal Depression in Rural India.” We sincerely appreciate the valuable feedback from you and the reviewers, which has greatly improved our work. As suggested, we focused on your comments and those raised by editor. We have addressed all the comments point-by-point and carefully revised the manuscript.

Editor 2 Comments#

Comment 1

However, conclusion part needs revision with crisp message of the study and any programmetic and/or policy direction.

Response

“Mental health and gender equality, central to SDGs 3 and 5, remain under-addressed in India’s primary healthcare system. This study from rural Chatra district, Jharkhand, reveals a high prevalence of antenatal and postnatal depression and strong links with adverse birth outcomes. Key predictors of antenatal depression include intimate partner violence, higher gravidity, caste disadvantage, nuclear family living, and mass media exposure, while negative family reactions, adverse birth experiences, and multiple deliveries increase postnatal depression risk. Women from Scheduled Castes and Tribes face disproportionate vulnerability due to intersecting structural, cultural, and socioeconomic barriers, including stigma, poverty, limited services, gender-based violence, and restricted autonomy. Maternal healthcare must ensure that every woman is treated with dignity and equity, irrespective of caste, socioeconomic status, or family context.

The findings highlight the urgent need to integrate culturally sensitive mental health screening, IPV identification, and family-centred counselling into routine antenatal and postnatal care, particularly through community platforms such as ASHA-led services. Strengthening referral linkages between primary and specialized mental health care is essential to prevent intergenerational impacts. Leveraging existing maternal health programs to deliver scalable, community-based mental health interventions can substantially improve outcomes for mothers and children in rural India. (page 25-26, lines 433-449).”

---

## [Editor Report · Decision Letter 2]

18 Feb 2026

Antenatal Depression and Its Relationship with Birth Outcomes and Postnatal Depression in Rural India: A Longitudinal Study

PONE-D-25-16317R2

Dear Mr. Deepak,

We’re pleased to inform you that your manuscript has been judged scientifically suitable for publication and will be formally accepted for publication once it meets all outstanding technical requirements.

Kind regards,

Rajesh Raushan, PhD

Academic Editor

PLOS One
---

## [Editor Report · Acceptance letter]

PONE-D-25-16317R2

PLOS One

Dear Dr. -,

I'm pleased to inform you that your manuscript has been deemed suitable for publication in PLOS One. Congratulations! Your manuscript is now being handed over to our production team.

Kind regards,

on behalf of

Dr. Rajesh Raushan

Academic Editor

PLOS One